# One-Year Compliance After Calcitonin Gene-Related Peptide Monoclonal Antibody Therapy for Migraine Patients in a Real-World Setting: A Multicenter Cross-Sectional Study

**DOI:** 10.3390/jcm14030734

**Published:** 2025-01-23

**Authors:** Mi-kyoung Kang, Jong-Hee Sohn, Myoung-Jin Cha, Yoo Hwan Kim, Yooha Hong, Hee-Jin Im, Soo-Jin Cho

**Affiliations:** 1Department of Neurology, Dongtan Sacred Heart Hospital, College of Medicine, Hallym University, Hwaseong 18450, Republic of Korea; twomk4960@hallym.or.kr (M.-k.K.); dbgk486@naver.com (Y.H.); coolere@naver.com (H.-J.I.); 2Department of Neurology, Chuncheon Sacred Heart Hospital, College of Medicine, Hallym University, Chuncheon 200704, Republic of Korea; deepfoci@hallym.or.kr; 3Department of Neurology, National Police Hospital, Seoul 05715, Republic of Korea; mjccja@police.go.kr; 4Department of Neurology, Hallym University Sacred Heart Hospital, College of Medicine, Hallym University, Anyang 14068, Republic of Korea; drneuroneo@gmail.com

**Keywords:** anti-CGRP monoclonal antibody, real-world data, compliance

## Abstract

**Background:** Calcitonin gene-related peptide monoclonal antibodies (CGRP mAbs) are a breakthrough migraine treatment, but long-term compliance under limited public insurance coverage has not been well known. This study explores one-year treatment patterns and outcomes of CGRP mAbs using real-world data. **Methods:** This multicenter retrospective study included migraine patients treated with CGRP monoclonal antibodies (CGRP mAbs) from July 2022 to June 2023. Treatment discontinuation was defined as a gap of over 60 days between injections. Among patients with 12 months of follow-up, adherence was measured using the Proportion of Days Covered (PDC), calculated as the ratio of days covered to the follow-up duration, with PDC ≥ 80% indicating good adherence. Efficacy was also assessed, defined as a ≥50% reduction in monthly headache days and acute medication use. **Results:** The study included 140 patients (mean age 44.6 ± 12.1 years; 82.9% female). Migraine without aura was predominant (93.6%), and 65.0% had chronic migraine. CGRP mAbs discontinuation occurred in 71.4% of patients, primarily due to headache improvement (22.9%) or lack of efficacy (15.0%). Among 81 patients with 12 months of follow-up, good adherence was observed in 40.7% of patients. Among these patients, 60.6% achieved a ≥50% reduction in monthly headache days, and 51.9% showed a ≥50% reduction in monthly acute medication use. **Conclusions:** More than two-thirds of patients discontinued the CGRP mAb within 1 year, so these findings emphasize the need for strategies to improve adherence and optimize follow-up plans to enhance patient support.

## 1. Introduction

Migraine is a neurological disorder that affects millions of people worldwide, significantly impairing quality of life and contributing to a substantial economic burden on healthcare systems [1]. Despite its high prevalence, effective preventive treatments have remained a challenge, particularly for patients with chronic or refractory migraine. Recent studies have identified calcitonin gene-related peptide (CGRP) as a central mediator in the pathogenesis of migraine, playing a key role in pain signaling and neurogenic inflammation [2,3,4]. The recognition of the role of CGRP in migraine has paved the way for the development of CGRP monoclonal antibodies (mAbs), a preventive therapy specifically designed to modulate the CGRP pathway. Currently, CGRP mAbs, such as fremanezumab, galcanezumab, and erenumab, are widely used in clinical practice [5]. In South Korea, fremanezumab and galcanezumab are the primary CGRP mAbs available for migraine prevention.

The latest migraine treatment guidelines from the American Headache Society [6], the European Headache Federation (EHF) [7], and the Korean Headache Society [8] recommend CGRP mAbs as essential treatment options for migraine patients. And EHF recently issued a consensus statement to include CGRP pathway preventive drugs in first-line preventive therapy, and long-term and real-world data confirm their sustained efficacy and safety. The guidelines recommend the following dosing regimens for CGRP mAbs: erenumab and galcanezumab are given monthly, while fremanezumab can be given monthly or quarterly [7]. Treatment typically continues for 6 to 12 months, after which the patient’s condition and response are assessed to determine whether therapy should be continued [9].

CGRP mAbs require regular injections and long-term treatment, making patient compliance essential for sustained therapeutic success. However, factors such as migraine control, insufficient response, adverse reactions, pregnancy, financial constraints, or personal preference often led to treatment discontinuation. While most studies focus on the efficacy and tolerability of CGRP-targeted therapies [2,10,11], understanding of their use in the real world remains limited. In particular, there is a scarcity of research examining the long-term use of CGRP mAbs in clinical practice, especially with data extending beyond 6 months. Real-world data are essential for translating clinical trial findings into everyday medical practice, reflecting the complexities and challenges faced by diverse patient populations in different healthcare systems. This study aims to analyze the treatment patterns of CGRP mAbs and assess compliance for up to one year in real-world migraine management, providing practical data on usage patterns.

## 2. Materials and Methods

### 2.1. Study Design and Participants

This multicenter, retrospective observational study was conducted in four centers and was reviewed and approved by the Institutional Review Board of Dongtan Hallym University Sacred Heart Hospital, Republic of Korea (approval number: [HDT 2024-08-010-001] on 1 February 2023). All study procedures adhered to the principles of the Declaration of Helsinki.

The inclusion criteria were adult patients aged 18 years or with a diagnosis of migraine according to the International Classification of Headache Disorders, 3rd edition [12], who received CGRP mAb treatment between July 2022 and June 2023.

### 2.2. Data Collection

Patient demographics, including age, gender distribution, migraine subtypes (such as migraine with or without aura, chronic migraine, pure menstrual migraine, and menstrual-related migraine), age at migraine onset, and disease duration prior to CGRP mAb treatment initiation, were collected. Data on CGRP mAb type (galcanezumab or fremanezumab), treatment duration, number of injections, injection intervals, and observation periods were also collected and analyzed. In addition, treatment patterns, including discontinuation and adherence, were assessed.

### 2.3. Treatment Pattern Analysis

Participants were followed for up to one year after their initial injection of CGRP mAbs, which was administered between July 2022 and June 2023. During this time, detailed analyses of treatment patterns were conducted, focusing on three key aspects: treatment discontinuation, defined as patients stopping treatment; adherence, which measured the consistency of treatment over the follow-up period; and efficacy according to adherence level.

Treatment discontinuation was defined as patients on CGRP mAb therapy not receiving their next injection within 60 days of the previous injection for monthly regimens (twice the standard dosing interval). For patients receiving injections every three months, discontinuation was defined as not receiving the next injection within 120 days of the previous injection. These criteria were based on the standard dosing schedules specified in the drug’s prescribing information [13]. And reasons for treatment discontinuation were assessed and recorded at the time of discontinuation during the follow-up period.

For patients who completed the full 12-month follow-up, adherence was assessed using the Proportion of Days Covered (PDC) method, which is a widely accepted measure in retrospective studies to assess medication adherence [14,15]. PDC is calculated by dividing the total number of days the medication was available to the patient by the total number of days in the follow-up period (365 days for 12 months). Each injection of CGRP mAbs was assumed to provide coverage for 30 days of coverage for monthly regimens and 90 days for quarterly regimens from the date of the prescription. Patients with a PDC of ≥80% were classified as adherent, as this threshold is commonly applied in adherence studies to indicate good medication adherence [13,14,16]. Additionally, we analyzed efficacy based on rates of ≥50% reduction in monthly headache days (MHDs) and monthly acute medication use (MAM) according to CGRP mAb type and PDC levels.

Missing data were excluded from the analysis to ensure accurate and reliable results. Given the retrospective nature of this study, a complete case analysis was conducted, including only patients with complete data on key variables such as adherence and treatment outcomes in the final analysis.

### 2.4. Statistical Analysis

All statistical analyses were performed using IBM SPSS Statistics 29.0.2.0. Descriptive statistics included demographic data and baseline disease characteristics. Continuous data are presented as mean and standard deviation (SD) or median and interquartile range (IQR), and categorical data are presented as frequencies and percentages. Adherence levels were divided into four categories, and the proportion of patients achieving a ≥50% reduction in monthly headache days and monthly acute medication days was analyzed within each category. Group comparisons for categorical variables, including adherence levels and CGRP type, were conducted using the chi-square test. Fisher’s exact test was used to compare the proportion of patients achieving a ≥50% reduction in monthly headache days and monthly acute medication days across different PDC levels. This test was chosen due to the small sample size in some categories, which did not meet the assumptions required for the chi-square test. A *p*-value of less than 0.05 was considered statistically significant.

## 3. Results

### 3.1. Patient Demographics

The study population consisted of 140 patients with a mean age of 44.6 years (±12.1), of whom 82.9% were female (116/140) and 17.1% were male (24/140). The mean body mass index (BMI) was 22.2 (±3.4). Comorbidities included hypertension in 12.1% (17/140) of patients and diabetes mellitus in 2.1% (3/140).

Patient demographic details are presented in Table 1. Migraine without aura was observed in 93.6% (131/140) of patients, while 6.4% (9/140) had migraine with aura. Of the patients, 65.0% (91/140) had chronic migraine, 5.0% (7/140) had pure menstrual migraine, and 8.6% (12/140) had menstrual-related migraine. The mean age of migraine onset was 26.6 years (±12.6), and the mean disease duration before initiating CGRP mAb therapy was 16.2 years (±12.0). Regarding the type of anti-CGRP mAb, 41.4% (58/140) received galcanezumab, and 58.6% (82/140) received fremanezumab (Table 1). A total of 28.6% (40/140) of patients maintained anti-CGRP mAb treatment for 12 months. The median number of days of anti-CGRP mAb treatment was 179.5 (IQR: 93.0–360.0). The median number of injections was 5.0 (IQR: 3.0–9.0), with 17.1% (24/140) receiving three injections, being the most common frequency (Figure 1). The median interval between injections was 32.0 days (IQR: 30.0–36.0). In addition, 51.4% (72/140) of patients were adherent, with a PDC of 80% or higher.

### 3.2. Discontinuation

During the 12-month follow-up period, 71.4% (100/140) of patients discontinued treatment (Table 1). Among these, 59 patients discontinued treatment and dropped out completely, while 41 patients discontinued treatment but were followed for the remainder of the 12 months. Of these, nine patients resumed treatment after discontinuation. Five resumed treatments with the same type of CGRP mAb, including one who initially started with galcanezumab and four with fremanezumab. Four patients switched to a different type when restarting treatment: two transitioned from galcanezumab to fremanezumab, and two switched from fremanezumab to galcanezumab.

The most common reason for discontinuation was headache improvement, accounting for 22.9% (32/140) of cases. Other reasons included lack of efficacy (15.0%, 21/140), side effects (5.0%, 7/140), financial difficulties (2.1%, 3/140), end of insurance coverage (1.4%, 2/140), and pregnancy planning (1.4%, 2/140). The side effects leading to treatment discontinuation included allergic reactions in three patients (2.1%), weight gain in two patients (1.4%), constipation in one patient (0.7%), and abnormal uterine bleeding in one patient (0.7%) (Figure 2).

### 3.3. Adherence

We conducted an analysis of the adherence to CGRP mAbs among patients who had completed the full 12-month follow-up. Of the 140 patients, 81 completed the 12-month follow-up period (Figure 2), with a mean PDC of 61.6% (IQR: 34.4–996.7). At the 12 months after treatment initiation, 40.7% (33/81) of patients were considered adherent, with 37.0% (10/28) in galcanezumab users and 43.4% (23/53) of fremanezumab users. Furthermore, 21.0% (17/81) exhibited a PDC between 50–79%, 17.3% (14/81) demonstrated a PDC between 30–49%, and 21.0% (17/81) presented with a PDC of less than 30% (Figure 3A).

### 3.4. Comparison of Efficacy According to PDC

The efficacy of CGRP mAbs was assessed by analyzing the percentage of patients who achieved a ≥50% reduction in MHD across different adherence levels. Data on MHD were collected from 79 patients, and the results showed that 60.6% (20/33) with a PDC ≥ 80% achieved a ≥50% reduction in MHD. The proportions of patients who achieved a ≥50% reduction in MHD were depicted by PDC and types of CGRP mAbs. No statistically significant differences were observed in MHD improvements across PDC levels or between galcanezumab and fremanezumab (*p* = 0.769, Figure 3B).

We also assessed the proportion of patients achieving a ≥50% reduction in MAM use across PDC levels. Data on MAM use were collected from 64 patients, and the results were as follows: overall, 51.9% (14/27) of patients with a PDC ≥ 80% achieved a ≥50% reduction in MAM use. The proportions of patients who achieved a ≥50% reduction in MAM use were depicted by PDC and types of CGRP mAbs. There were no significant differences in MAM reduction across PDC levels and between the two medications (*p* = 0.927, Figure 3C).

## 4. Discussion

In this study, we analyzed treatment patterns and compliance of CGRP mAbs in migraine patients in real-world settings. During the 12-month period, 71.4% of patients discontinued treatment, most commonly due to headache improvement (22.9%, 32/140). The median number of days of anti-CGRP mAb treatment was 179.5 (IQR: 93.0–360.0). The median number of injections was five (IQR: 3.0–9.0). Among patients with 12 months of follow-up, 40.7% were adherent with PDC ≥ 80%. At 1 year, a ≥50% reduction in MHD and MAM, respectively, was achieved in 60.5% and 51.9% of patients with a PDC ≥ 80%. There were no significant differences in MHD or MAM reduction across PDC levels and between the two medications.

Approximately 71.4% of patients discontinued CGRP mAb treatment within 12 months, which is higher than 58.8% reported in previously claimed data and similar to 26.7% of persistence rates among patients with chronic migraine studies [16,17]. The reasons for discontinuation were varied; the most common was headache improvement. While only 2.1% of patients cited financial constraints as their primary concern, it is possible that financial burden may influence CGRP mAb initiation rather than discontinuation. Improving insurance support and reimbursement policies may reduce financial barriers and increase access to CGRP mAb therapy, potentially leading to better long-term treatment maintenance.

In this study, the duration of treatment and the number of injections exhibited variability, with a median CGRP mAb treatment duration of 179.5 days and a median of five injections during the observation period. In comparison, previous studies [18,19,20], including one reporting a mean treatment duration of approximately 252.3 days and a mean of seven injections [16], demonstrated slightly longer treatment durations and more injections. The findings underscore the challenges of maintaining long-term consistency in clinical practice within real-world settings and emphasize the importance of developing personalized treatment strategies with variable dosing intervals to accommodate patient-specific needs and improve treatment maintenance, particularly in settings with limited coverage.

An analysis of adherence among patients who completed the full 12-month follow-up revealed that 40.7% were considered adherent (PDC ≥ 80%). This finding is consistent with a previous study of galcanezumab users, which reported an adherence rate of 44.2% [16]. However, it is lower than the 87% adherence reported in Ireland with reimbursement, which may be attributable to differences in patient selection criteria and healthcare support systems [13]. Earlier studies have shown that adherence tends to be higher in the context of managed access programs (MAPs), where strict eligibility criteria ensure that only treatment-resistant patients receive CGRP mAbs [13,21,22,23]. These programs often provide additional support and follow-up, contributing to improved adherence. By contrast, in real-world clinical practice, adherence may be influenced by factors such as insurance limitations, inconsistent follow-up practices, and individual patient circumstances, which may not offer the same level of structured support [24,25].

There were no significant differences in MHD or MAM reduction across PDC levels and between the two medications. These findings may be related to the proportion of discontinuation with headache improvement. Considering the roller coaster clinical course of migraine, some patients may benefit from monthly dosing, while others might achieve sufficient control with less frequent injections [26,27].

Our study has several limitations. As a retrospective observational nature, it was challenging to capture detailed personal factors that might influence treatment discontinuation and adherence, limiting our ability to establish cause-and-effect relationships. The study’s sample size is relatively small (140 patients), and the findings are specific to a single country’s healthcare system (South Korea). Variations in healthcare systems, insurance coverage, and reimbursement policies across regions may limit the study’s ability to draw conclusions that are applicable to other populations. The influence of concomitant preventive medications, which some patients may have been using, was not fully accounted for, and this could affect outcomes, as these medications may alter the perceived efficacy and adherence to CGRP mAb therapy. Additionally, the lack of data on patients lost to follow-up may have further impacted the interpretation of adherence and outcomes.

Despite these limitations, our study provides valuable real-world insights into the treatment patterns and adherence of CGRP mAbs within a unique healthcare setting. It is noteworthy that this is the first study conducted in South Korea to evaluate the real-world use of CGRP mAbs, providing a localized perspective on the challenges associated with the maintenance of long-term therapy within a healthcare system with limited insurance coverage. By identifying key reasons for treatment discontinuation, such as lack of follow-up and financial constraints, this study contributes to a better understanding of barriers to sustained treatment. The comprehensive evaluation of clinical outcomes across adherence levels underscores the importance of developing tailored strategies to optimize treatment success.

Future research with larger, multi-country cohorts and consideration of concurrent medication use is needed to improve sample diversity, enhance generalizability, and validate and expand upon these findings. Additionally, comparative studies between CGRP mAbs and other migraine treatments are essential to better contextualize their efficacy and adherence. Evaluating the cost-effectiveness of CGRP mAbs, particularly in healthcare systems with limited public insurance coverage, is also crucial to provide practical insights into their long-term use and guide healthcare policy decisions.

## 5. Conclusions

In this real-world study, 71.4% of patients discontinued CGRP mAb treatment within 1 year, reflecting the difficulty of maintaining long-term therapy, particularly within limited healthcare systems. The most common reason for discontinuation was headache improvement, and MHD or MAM reduction for 1 year was similar across PDC levels. These findings highlight the need for strategies to improve adherence and optimize follow-up plans to enhance patient support.

## Figures and Tables

**Figure 1 jcm-14-00734-f001:**
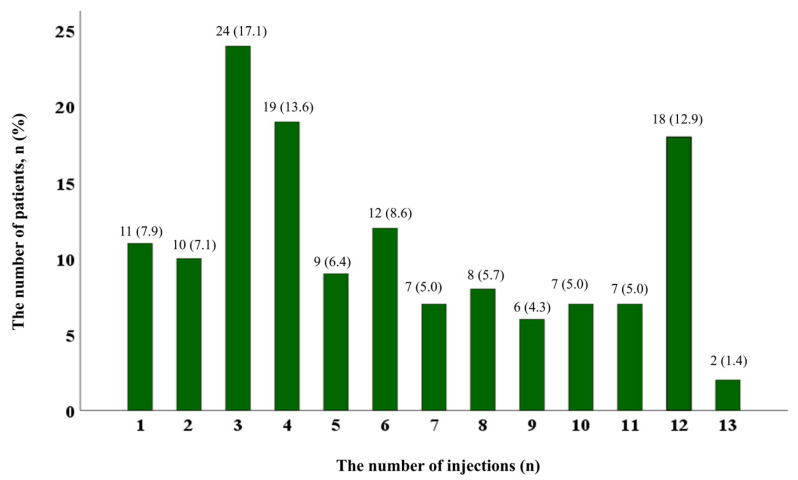
Distribution of the number of injections of the total 140 patients. Distribution of patients by the total number of injections received during the 12-month follow-up period. The most common number of injections was 3 (17.1%), followed by 4 (13.6%) and 12 (12.9%) injections.

**Figure 2 jcm-14-00734-f002:**
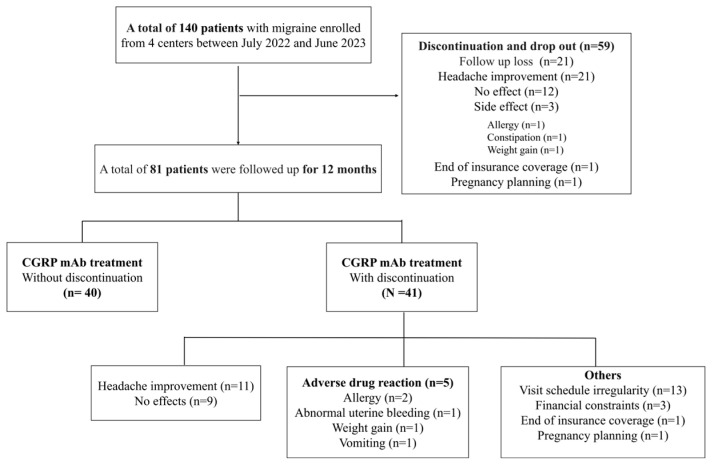
Flowchart of patient enrollment and reasons for discontinuation of anti-CGRP mAb treatment in four centres. Reasons for treatment discontinuation were identified based on multiple response options.

**Figure 3 jcm-14-00734-f003:**
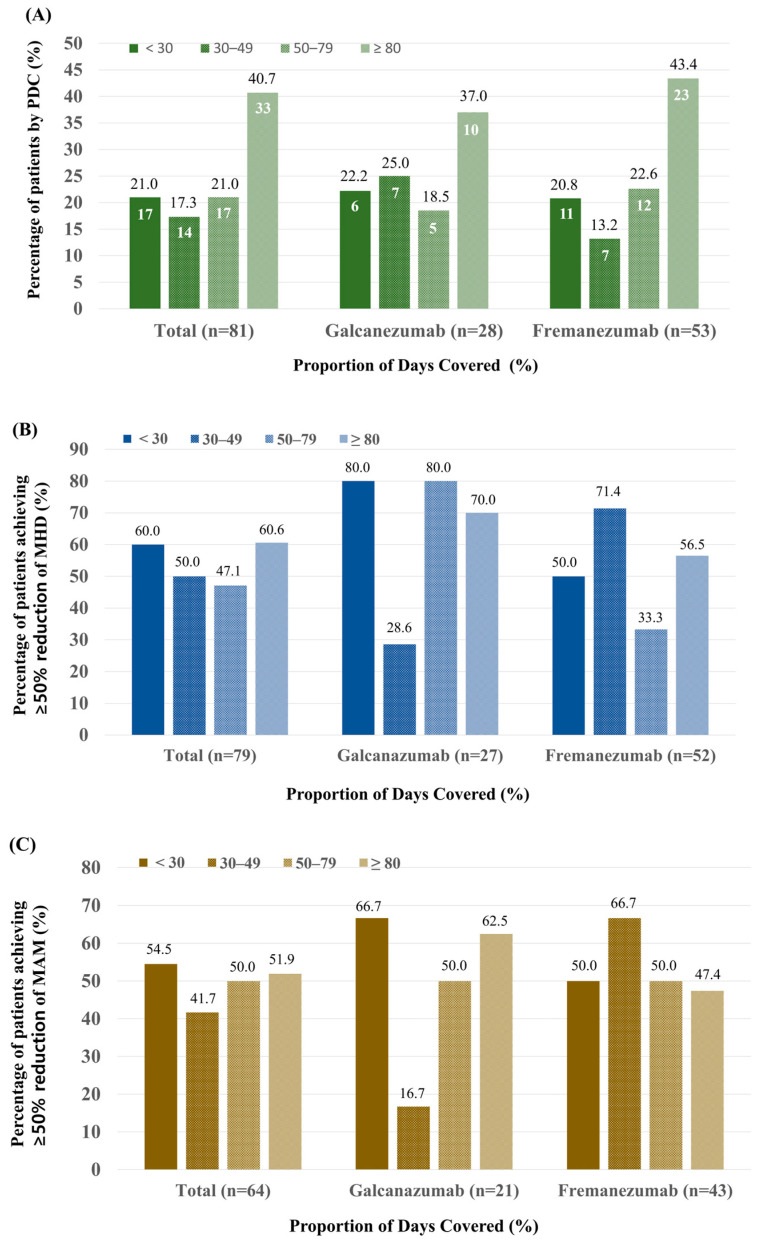
Proportion of Days Covered (PDC) among 81 patients treated with anti-CGRP mAbs over 12 months and clinical efficacy based on PDC. (**A**) Distribution of patients based on adherence levels measured by the PDC. Among patients who completed 12 months of follow-up, 40.7% achieved a PDC of ≥80%, indicating high adherence, while 21.0% had a PDC of <30%, indicating low adherence. (**B**) Percentage of patients achieving a ≥50% reduction in monthly headache days. MHD stratified by PDC levels (<30%, 30–49%, 50–79%, and ≥80%) and by CGRP mAb type (galcanezumab and fremanezumab). (**C**) Percentage of patients achieving a ≥50% reduction in monthly acute medication use (MAM) stratified by PDC levels and CGRP mAb type. The number inside the bar in the figure (**A**) represents the number of patients, and the number outside the bar represents the percentage of patients.

**Table 1 jcm-14-00734-t001:** Clinical characteristics of patients (*n* = 140).

Age, mean (SD)	44.6 (±12.1)
Sex, *n* (%)	
female	116 (82.9)
male	24 (17.1)
BMI, mean (SD)	22.2 (±3.4)
Hypertension, *n* (%)	17 (12.1)
DM, *n* (%)	3 (2.1)
Migraine subtype, *n* (%)	
Migraine without aura	131 (93.6)
Migraine with aura	9 (6.4)
Chronic migraine	91 (65.0)
PMM	7 (5.0)
MRM	12 (8.6)
Age of onset, years old, mean (SD)	26.6 (±12.6)
Disease duration before CGRP, year, mean (SD)	16.2 (±12.0)
Anti-CGRP mAbs type, *n* (%)	
Galcanezumab	58 (41.4)
Change to fremanezumab	8 (5.7)
Fremanezumab	82 (58.6)
Change to galcanezumab	8 (5.7)
Anti-CGRP continue for 12 months, *n* (%) *	40 (28.6)
The number of days of anti-CGRP treatment, median (IQR)	179.5 (93.0–360.0)
Number of injections, median (IQR)	5.0 (3.0–9.0)
Adherence, *n* (%)	
<30%	20 (14.3)
30–49%	22 (15.7)
50–79%	26 (18.6)
≥80%	72 (51.4)
Intervals of injections, days, median (IQR)	32.0 (30.0–36.0)

SD, standard deviation; DM, diabetes mellites; PMM, pure menstrual migraine; MRM, menstrual-related migraine; anti-CGRP mAbs, anti-calcitonin gene-related peptide monoclonal antibodies; IQR, interquartile range. *, including patients with temporary discontinuation or irregular schedules.

## Data Availability

The data supporting the findings of this study are available from the corresponding author upon reasonable request.

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
