# Peer review of "One-Year Compliance After Calcitonin Gene-Related Peptide Monoclonal Antibody Therapy for Migraine Patients in a Real-World Setting: A Multicenter Cross-Sectional Study"

_jcm, 2025, doi:10.3390/jcm14030734_

Round 1

Reviewer 1 Report

Comments and Suggestions for Authors

The authors in this study have used an estimate of proportion of days covered to address adherence and efficacy. The assumption is that this is because this is a retrospective study. It would have been good to have a brief description of this concept or at least a link to a useful article. 

The loss to follow-up was almost 30%. This is cited as the most common reason for discontinuation. Did the authors try and contact these patients to ascertain the reason? Among patients who completed the full 12-month follow-up about half in each MAB group remained adherent. One cannot necessarily make the presumption that a longer time to adherence results in better outcome without the full data set. E.g If those lost to follow up went into remission the assumption made is misleading. 

Paragraph 3.4 is not easy to follow although better represented in Figure B. It would be useful to have the actual numbers (n) with the percentages.  

It would be better to separate the data from the two drugs rather than pool the data. This is particularly relevant to the number of injections used unless all were monthly. The methods suggest this was not the case. 

The authors mention that the variability in persistence rates across studies may be influenced by differences in patient populations and healthcare systems. A number of studies have been cited. The authors can, from these studies, have a look and assess what is the difference between the cohorts which may account for the changes. 

The authors acknowledge that the sample was small, particularly as the main outcome was to look at compliance rates. If there had been more detail about outcomes it would be more useful. 

Reading through the paper was not easy to follow and in part this is because of the issues mentioned as above. 

Reviewer 2 Report

Comments and Suggestions for Authors

The study provides valuable real-world data on CGRP monoclonal antibody (mAb) therapy, especially within the South Korean healthcare setting. It examines compliance, persistence, and efficacy metrics over a 12-month period, offering insights into long-term treatment trends. The study benefits from clear inclusion criteria, data collection methods, and statistical analyses. This is also the first study of its kind in South Korea, addressing a unique healthcare context with limited insurance coverage. This study identified barriers to adherence and persistence, such as financial constraints and lack of follow-up, providing a foundation for policy and clinical recommendations.

There are a couple of points that the authors can address in the revised version:

The study focuses exclusively on a South Korean population, making it less applicable to global audiences with different healthcare systems. Please elaborate and encourage larger and worldwide studies.

The sample of 155 patients restricts statistical power and the ability to generalize findings to broader populations. This can be pointed out in the limitations and how to overcome the challenge. Propose multicenter collaborations across different countries to improve the sample diversity in the future studies.

The nature of the study design limits the ability to draw causal inferences regarding adherence and outcomes. At least the authors can comment on this point.

Factors like concurrent medications or comorbidities influencing adherence and efficacy were not fully analyzed.

Lack of detailed information on patient-reported outcomes (e.g., quality of life, satisfaction) limits the depth of the study's impact assessment.

Suggest that the next studies can compare CGRP mAbs with other migraine treatments to contextualize efficacy and adherence. In addition, propose to evaluate the cost-effectiveness of CGRP mAbs, particularly in settings with limited public insurance.

Use of digital tools (e.g., apps, e-health platforms) would be a good idea to improve follow-up and monitor adherence more effectively in future studies. The authors can also suggest to develop personalized treatment plans with variable dosing intervals to accommodate patient-specific needs.

Highlight findings to influence reimbursement and insurance policies, potentially reducing financial barriers.

This study has laid a robust foundation for understanding real-world compliance with CGRP mAb therapies. Addressing the outlined points and incorporating these improvements could significantly enhance the study's impact.

Reviewer 3 Report

Comments and Suggestions for Authors

Q1: Department of Neurology, Dongtan Sacred Heart Hospital, Hallym University “Colleage “?  "Colleage" should be corrected to "College."

Q2:Abstract: generally fine. Specify that this is a "retrospective" study in the methods section of the abstract to set clearer expectations

Q3: Introduction: provide a good background on CGRP monoclonal antibodies and their importance in migraine treatment. However, please focus more on the study's rationale and objectives rather than guidelines..

Q4: Methods: Well-defined criteria for persistence, adherence, and treatment discontinuation , but some details about statistical tests (e.g., why Fisher's exact test was used) could be expanded for clarity.

Q5 Methods: Please add justification for choosing the Proportion of Days Covered (PDC) ≥80% as the cutoff for good adherence.

Q6 Methods: Please add details about how missing data were handled, if applicable.

Q7 Methods: Please specify the time frame during which the reasons for discontinuation were recorded (e.g., at follow-up visits).

Q8: Results: Comprehensive demographic and clinical data presentation, but the use of multiple percentages in quick succession (e.g., in patient demographics) might overwhelm readers. Summarizing or using a table format earlier would improve clarity.

Q9:Figure 2 could benefit from clearer labels and a more reader-friendly design.

Q10: Explain why chi-square tests were used for adherence categories, and if assumptions were not met, describe adjustments.

Q11: Discussion:  Good synthesis of study findings and comparison with existing literature.

Q12: discussion:  Emphasize limitations (e.g., small sample size, observational design, observer bias) earlier in the discussion to provide appropriate context for the findings.

Q13: discussion: Discuss why adherence does not correlate significantly with efficacy.  How treatment adherence could be improved.

Q14: discussion: When comparing persistence rates with other studies, explicitly state why differences might exist (e.g., healthcare systems, cultural factors), and the potential impact of socio-economic factors beyond direct financial constraints.

Q15: Conclusion: The conclusion could briefly touch on how findings compare to global data or unique challenges in South Korea’s healthcare system. It will be more attractive to add a forward-looking statement about the need for tailored adherence strategies and policy-level changes to enhance treatment outcomes.

Round 2

Reviewer 3 Report

Comments and Suggestions for Authors

accepted